Unleashing the power of AI in detecting metal surface defects: an optimized YOLOv7-tiny model approach

Chen Shuaiting
Zhou Feng zfycit@ycit.edu.cn
Gao Gan
Ge Xiaole
Wang Rugang
Yancheng Institute of Technology , JiangSu , China
Shuja Junaid
Electronic publication date: 2024 Jan 22
Publication date: 2024
Volume: 10
Electronic Location ID: e1727
Received 2023 Aug 18; Accepted 2023 Nov 7
Copyright: ©2024 Chen et al.
Copyright year: 2024
Copyright holder: Chen et al.
License: This is an open access article distributed under the terms of the Creative Commons Attribution License, which permits unrestricted use, distribution, reproduction and adaptation in any medium and for any purpose provided that it is properly attributed. For attribution, the original author(s), title, publication source (PeerJ Computer Science) and either DOI or URL of the article must be cited.
License URL: https://creativecommons.org/licenses/by/4.0/

Keywords: Defect detection, YOLOv7-tiny, Deep learning, Metal surface, Feature fusion

Funding: Jiangsu Graduate Practical Innovation Project No. SJCX23_1873 Natural Science Research of Jiangsu Province Colleges and Universities 19KJA110002 Natural Science Foundation of China 61673108 Natural Science Research Project of Jiangsu Province Universities 18KJD510010 19KJB510061 Jiangsu Province Natural Science Foundation Project BK20181050 This work was supported by the Jiangsu Graduate Practical Innovation Project under grant numbers (No. SJCX23_1873), the major Project of Natural Science Research of Jiangsu Province Colleges and Universities under grant number (No. 19KJA110002), the Natural Science Foundation of China under grant number (No. 61673108), the Natural Science Research Project of Jiangsu Province Universities under grant number (No. 18KJD510010) and (No. 19KJB510061), the Jiangsu Province Natural Science Foundation Project under grant number (No. BK20181050). The funders had no role in study design, data collection and analysis, decision to publish, or preparation of the manuscript.

==============================
The detection of surface defects on metal products during the production process is crucial for ensuring high-quality products. These defects also lead to significant losses in the high-tech industry. To address the issues of slow detection speed and low accuracy in traditional metal surface defect detection, an improved algorithm based on the YOLOv7-tiny model is proposed. Firstly, to enhance the feature extraction and fusion capabilities of the model, the depth aware convolution module (DAC) is introduced to replace all ELAN-T modules in the network. Secondly, the AWFP-Add module is added after the Concat module in the network’s Head section to strengthen the network’s ability to adaptively distinguish the importance of different features. Finally, in order to expedite model convergence and alleviate the problem of imbalanced positive and negative samples in the study, a new loss function called Focal-SIoU is used to replace the original model’s CIoU loss function. To validate the effectiveness of the proposed model, two industrial metal surface defect datasets, GC10-DET and NEU-DET, were employed in our experiments. Experimental results demonstrate that the improved algorithm achieved detection frame rates exceeding 100 fps on both datasets. Furthermore, the enhanced model achieved an mAP of 81% on the GC10-DET dataset and 80.1% on the NEU-DET dataset. Compared to the original YOLOv7-tiny algorithm, this represents an increase in mAP of nearly 11% and 9.2%, respectively. Moreover, when compared to other novel algorithms, our improved model demonstrated enhanced detection accuracy and significantly improved detection speed. These results collectively indicate that our proposed enhanced model effectively fulfills the industry’s demand for rapid and efficient detection and recognition of metal surface defects.

Introduction

Metals, as essential industrial materials, are widely used in various sectors such as machinery, aerospace, automotive, defense, and light industries. However, factors such as raw material quality, production environment, equipment, and human errors often lead to surface defects in metal during the industrial production process. These defects include Crease, Water_spot, Punching_hole, Inclusion, etc. Moreover, products are easily damaged on the surface during the actual production process. To prevent the supply of such substandard products to other industries, which could cause unnecessary economic losses and harm to personnel, the detection of metal surface defects becomes a top priority (Kim et al., 2017). In response to this issue, scholars have been devoted to researching metal defect detection and identification technologies to improve the quality and efficiency of metal production and provide high-quality metal materials for various industrial applications.

In recent decades, scholars both domestically and internationally have been researching how to efficiently classify and detect defects. Traditional image processing techniques have been widely used for defect detection, in addition to manual visual inspection  (Li et al., 2018; Senthikumar, Palanisamy & Jaya, 2014). However, manual visual inspection requires workers to engage in long hours of intensive labor, which eventually leads to decreased identification quality and inefficient productivity. On the other hand, for traditional image processing technology, the use of specially designed manual features (such as: Local Binary Pattern (LBP) (Song & Yan, 2013), Histograms of Orientated Gradients (HOG) (Shumin, Zhoufeng & Chunlei, 2011), Gray-level Co-occurrence Matrix (GLCM) (Chondronasios, Popov & Jordanov, 2016) and other features (Gibert, Patel & Chellappa, 2016; Tao et al., 2017)) to classify surface defects is the most important link, the existence of this link improves the performance of surface defect detection to a certain extent. Nevertheless, due to the strong sensitivity of carefully designed features to factors such as lighting intensity and different backgrounds, the robustness and generalization ability of defect detection methods may be poor (Zhou et al., 2020). To address this limitation, deep learning methods have emerged, particularly convolutional neural networks (CNNs), which overcome the problem of manual feature extraction. CNNs automatically capture both shallow-level texture information and deep semantic features, the methods based on CNNs exhibit stronger robustness and generalization ability compared to traditional methods. Consequently, CNNs have become a crucial method in the industrial sector (Singh & Desai, 2023; Lui, Chan & Leung, 2021; Lui, Chan & Leung, 2022).

In order to accelerate industrial production, enhance product quality and optimize labor resource utilization, many researchers are committed to integrating deep learning target recognition methods into industrial manufacturing workflows. Current deep learning-based object detection algorithms are basically divided into two categories: two-stage algorithms and one-stage algorithms. Two-stage algorithms achieve higher detection accuracy. In the first stage, they use Region Proposal Networks (RPN) to predict region proposals, then retrieve features for each region of interest (ROI) through merging. These features are then fed into two branches of the network, one for object classification and the other for bounding box regression. Common two-stage object detection algorithms include R-CNN (Girshick et al., 2014), Fast R-CNN (Girshick, 2015), and Faster R-CNN (Ren et al., 2015). On the other hand, one-stage networks extract image features using multiple layers of convolutional neural networks and directly input them into the two branches of the network for defect classification and bounding box regression. Therefore, this type of algorithm has a faster detection speed but slightly lower detection accuracy, especially for small overlapping objects. Common one-stage object detection algorithms include SSD (Liu et al., 2016) and the YOLO series of algorithms (Redmon et al., 2016; Redmon & Farhadi, 2017; Redmon & Farhadi, 2018; Bochkovskiy, Wang & Liao, 2020).

The etiology of surface defects in metals is multifaceted, exhibiting a range of complex morphologies. Based on the geometric characteristics of these defects, they can generally be categorized into three main types: point-like, linear, and planar. Typical defects can be summarized as punching hole, welding line, crescent gap, water spot, oil spot, silk spot, inclusion, rolled pit, crease, and waist folding. Notably, water spot and oil spot are defects that usually present low contrast, making them easily confounded with other types of defects; water spot are especially prone to misidentification. Given the diverse and intricate nature of these defect types, failure to adequately detect them during the production and processing of metals can have incalculable adverse effects on the structural integrity and functionality of the metal products. Consequently, the task of accurately identifying surface defects in metals is both highly necessary and critically important. However, there are still limitations in defect detection tasks, as directly using existing object detection models may not effectively detect certain types of defects. For metal surface defect detection, the characteristics of deep metal surfaces pose some specific challenges to the detection task. The first challenge is the variation in defect shape and size. Some types of metal surface defects are too small, while others are too large, and there is significant intra-class variation among defects. These obstacles make it difficult to detect small defects, while relatively larger defects are also challenging to detect due to variations in the shape of the same type of defect. To address this challenge, it is necessary to process and extract effective feature information from numerous defects of different scales, as well as improve the robustness of the detection model. The second challenge lies in detection efficiency, which is a critical aspect for industrial applications. It is imperative to maximize the model’s detection accuracy while meeting the requirements for real-time detection. Low detection speeds can lead to bottlenecks in the production line, thereby slowing down the manufacturing process, increasing costs, and necessitating expensive hardware and substantial computational resources. Conversely, low detection accuracy results in elevated false positive rates, where non-defective regions are inaccurately flagged as defective. This substantially increases the risk of missing small or hard-to-detect defects, consequently undermining the reliability of quality control measures. Hence, achieving efficient and accurate identification of surface defects in metals presents a formidable challenge.

To solve the above-mentioned problems, this study proposes an improved method for detecting metal surface defects based on YOLOv7-tiny. The method can deploy a high-speed and accurate model in the production process to achieve unmanned, fast, and accurate localization and classification of metal surface defects.

The main contributions of this study are as follows: (1) An improved metal surface defect detection model based on YOLOv7-tiny is proposed. (2) In order to further enhance the network’s ability to extract defect features and the interaction between spatial and channel information, the Elan-T modules in the YOLOv7-tiny model were replaced with the DAC modules tailored to this specific dataset, based on experimental retrieval. (3) To enable the network to automatically determine the importance of features and increase the ability of feature fusion, an Adaptive Weighted Feature Path module (AWFP-Add) is proposed. (4) To address the slow convergence speed, inaccurate regression results, and the imbalance problem in boundary box regression that is generally ignored by traditional loss functions, a new loss function called Focal-SIOU loss is used to replace the original boundary box loss function of YOLOv7-tiny.

Related Work

Surface defect detection is an important link in the metal production process. However, there are great differences in the defect scale of defects in the defect detection process. In order to improve the detection accuracy, Beskopylny et al. (2023) proposed a feature extraction network that utilizes depth-wise separable convolution to enhance detection. They also introduced dilated convolutions in the spatial pyramid pooling (SPPF) module to enlarge the receptive field and incorporate contextual information. Lastly, they introduced a novel attention mechanism, namely the Multi-scale Enhanced Context Attention (MECA), to facilitate the extraction of multi-scale detailed information. The results demonstrate a 6.5% improvement in mean average precision(mAP) and a 5.75% improvement in F1 score compared to the original model. Chen et al. (2023) introduced the coordinate attention (CA) mechanism module to replace the spatial pyramid pooling (SPP) structure in YOLOX’s Backbone. They also proposed a novel EDE block to capture the complete features of surface defects. Finally, they addressed the low contrast issue of steel surface defect images by introducing the CLAHE data augmentation method. The best model achieved an accuracy of 82.6% at a frame rate of 100.2fps on the NEU-DET dataset. Cheng & Yu (2020) proposed RetinaNet with differential channel attention and adaptive spatial feature fusion. The results demonstrated that the new network achieved a 78.25 mAP, exhibiting a 2.92% improvement compared to RetinaNet. Xing & Jia (2021) proposed a convolutional network classification model with symmetric modules for feature extraction, and designed an optimized IOU (XIoU). The results demonstrate that their model achieved a mAP of 79.89% on NEU-DET and 78.44% on the self-made detection dataset. Yu, Cheng & Li (2021) proposed a new detection model called CABF-FCOS, which is based on the anchor-free approach. This deep network utilizes channel attention mechanism (CAM) attention mechanism and bidirectional feature fusion network (BFFN) for bidirectional feature fusion, enabling the identification of specific categories and precise locations of steel defects. The experimental results showed that the new network achieved an mAP of 76.68% on the NEU-DET dataset, an improvement of 4.43% compared to FCOS. Tang et al. (2023) introduced the Transformer structure as an alternative to the commonly used CNN network architecture. They utilized the self-attention mechanism to capture global information and employed parallel computation to enhance computational efficiency. Through the Swin Transformer, they extracted multi-scale features from the images. The combination of FPN and RPN facilitated the integration of features from different scales. Finally, they improved the ROI head to obtain the category and precise localization of defects. The achieved detection accuracy of the final model was 81.1%, surpassing numerous classical CNN-based detection methods.

To meet the current industrial production needs, researchers are increasingly studying how to continuously improve the accuracy of defect detection while ensuring real-time monitoring. This type of research primarily focuses on single-stage object detection networks, particularly the YOLO series algorithms. Liu & Ma (2023) improved the utilization of defect features by adjusting the receptive field at different scales and attention weight preferences through the addition of an expanded and weighted cross-stage feature pyramid network in the Neck. They maximized the extraction of useful information by enhancing the cross-stage partial connection with ResNet in the Backbone. In order to increase robustness, the Head section adopted a decoupled head. As a result, their algorithm achieved better detection results with 79.93% and 72.76% mAP on the GC10-DET and NEU-DET datasets, respectively. Liu & Jia (2023) proposed a new model, ST-YOLO, for detecting defects in steel. This model utilizes a streamlined fusion network structure to meet the computational requirements for classification and localization tasks. To optimize label assignment, a self-adjusting label assignment algorithm is introduced, which guides the model to flexibly complete training. This method achieves an average detection accuracy of 80.3% at a frame rate of 46 frames per second. Furthermore, it has demonstrated excellent performance in real defect detection applications. Zhang et al. (2023) introduced and optimized the weighted bidirectional feature pyramid network with embedded residual module in YOLOv5s, and preprocessed the images using Laplacian sharpening. The best model achieved an mAP of 86.8% on the NEU-DET dataset, while efficiently processing RGB images of size 640 × 640 at a speed of 51 FPS. Liu et al. (2022) enhanced the representation of dense small defects in YOLOv3’s DarkNet53 backbone network by adding an extra scale prediction layer on top of the existing three layers. They further improved the capability by densely linking multi-scale feature maps across layers. This method achieved an average detection accuracy of 89.24% and demonstrated the ability to detect nearly 26 images of size 416*416 pixels per second. Xu et al. (2023) developed an end-to-end steel surface defect detection and size measurement system based on YOLOv5. They employed BiFPN in the Neck section to enhance the feature fusion and introduced the CA mechanism in the Head section to strengthen the spatial correlation of steel surface. Furthermore, they proposed an adaptive anchor box generation method based on defect shape difference features. As a result, the improved YOLOv5 achieved high detection accuracy of 93.6% at a fast detection speed of 133 FPS. It also exhibited remarkable accuracy in locating small defective objects. In conclusion, the YOLO-based object detection algorithm effectively addresses the problem of surface defect detection, but requires a trade-off between speed and accuracy. Building on this research, this paper proposes an improved metal surface defect detection model based on YOLOv7-tiny. From an optimization standpoint, our solution offers a way to detect metal surface defects, and the excellent performance of the improved YOLOv7-tiny is demonstrated.

Method

The YOLOv7-tiny algorithm is a simplified version of Yolov7, which retains the model scaling strategy based on the cascade idea. It also improves the efficient long aggregation network (ELAN) for higher detection accuracy, with smaller parameter sizes and faster detection speeds. The main differences between them lie in the internal components and the depth and width of these components. The backbone of the YOLOv7-tiny network primarily employs the streamlined efficient long aggregation network (ELAN-T), MP structure, and Silu activation function. These choices result in relatively superior performance in feature extraction and detection speed.

The process of defect detection using the YOLOv7-tiny model (Fig. 1) is as follows:

Figure 1 YOLOv7-tiny model training process.

1. Firstly, load the relevant dataset based on the configuration file.

2. Next, preprocess the dataset to meet the input requirements of the YOLOv7-tiny model.

3. Input the preprocessed data into the YOLOv7-tiny model and start the iterative training process, updating the parameter values. Finally, when the model training reaches the set number of iterations, the network outputs the final model, signifying the end of the training process.

Improved network structure based on YOLOv7-tiny model

The improved YOLOv7-tiny algorithmic network structure architecture proposed in this study is marked in the box in Fig. 2, and detailed information is given in the subsequent three sections.

Figure 2 The architecture of the proposed defect detection network.

In this study, specific improvements in the YOLOv7-tiny algorithm structure are highlighted in Fig. 2 using colored boxes. The addition of the AWFP-add module in a light purple box strengthens the model’s feature extraction and cross-scale fusion abilities within the Head network. The replacement of the ELAN-T structure with the DAC module in light blue further enhances the feature extraction capabilities of YOLOv7-tiny. Finally, the regression loss of the bounding boxes is substituted with the Focal-SIOU loss function, improving the accuracy of the regression results and expediting convergence.

Depth Aware Convolution module

In the YOLOv7-tiny network, the ELAN-T module is used to extract features from feature maps of various regions. However, for our metal surface defect dataset, defects within the same category vary greatly in size and shape. Furthermore, there may also be similarities between defects of different categories. Moreover, due to the variations in different sample materials and the influence of lighting conditions, the grayscale values of intra-class defect images also undergo certain changes, which in turn disrupt the accuracy of the detection results. These factors collectively make it challenging for the network to extract meaningful features. The ELAN-T module is a simplified version of the ELAN module in YOLOv7. However, the ELAN module itself is not very effective in feature extraction (Wang et al., 2023b). Hence, it is clear that the feature extraction capability of ELAN-T is also unsatisfactory. Therefore, we attempt to improve the ELAN-T module of YOLOv7-tiny by introducing our innovative depth aware convolution module (DAC) (as shown in Fig. 3). We replace all ELAN-T modules in the entire model structure with DAC modules to further enhance the defect feature extraction and fusion capabilities in the YOLOv7-tiny network architecture while maintaining a certain inference speed.

Figure 3 Improved C5 module (DAC module).

(A) ELAN-T module. (B) DAC module.

The inspiration for improving the DAC module comes from the combination of V7 and V7-tiny themselves. Due to the simple structure of the ELAN-T (C5) module in V7-tiny, its feature extraction capabilities are significantly limited compared to other models in the V7 series; The ELAN-T module consists of five convolutions: three 1 × 1 convolutions and two 3 × 3 convolutions. It is known that 1 × 1 convolutions only facilitate information exchange and feature fusion between channels, lacking interactions and fusion between neighboring pixels. On the other hand, there are only two 3 × 3 convolutions that enhance feature fusion between neighboring pixels. Consequently, it is reasonable to expect a poor feature extraction capability from the ELAN-T module. In summary, we enhance the C5 module by increasing the number of 1 × 1 and 3 × 3 convolutions. The 1 × 1 convolutions strengthen the information exchange among channels, while the increased 3 × 3 convolutions widen the receptive field, promoting better interaction of spatial information and feature fusion. Additionally, we employ the Concat operation to increase the number of features, consequently improving the non-linear transformation of the network.

AWFP-add module

YOLOv7-tiny incorporates the feature fusion network of the YOLOv5 series, which consists of the Feature Pyramid Network (FPN) and the Path Aggregation Network (PAN) architecture (Wang, Bochkovskiy & Liao, 2023a). Lin et al. (2017) effectively transfers strong semantic information from deeper feature layers to even deeper layers (Chen et al., 2021). On the other hand, PAN transmits accurate localization information from bottom to top. By combining FPN and PAN, different detection layers from the backbone are parameterized together, enhancing the feature fusion capability of the network. However, this combination introduces a drawback: the PAN structure takes as input the features that have been processed by the FPN, but some defect features extracted from the backbone’s original information are lost along the way. The lack of original information for learning can lead to bias in training and consequently affect detection accuracy. To address this issue, we propose incorporating the concepts of bi-directional weighted feature pyramid network (BiFPN) and fast normalization fusion from its associated paper into the conventional Add module. This allows the Add module to have its own learnable parameters. Additionally, in terms of structure, we insert the Add module after each of the three Concat modules in the feature fusion network. We also combine the practical significance of surface defect detection in this study and the concept of residuals. Specifically, we connect the Add module and Backbone to the three feature maps provided to the Head, enabling the network to retain more shallow semantic information without losing too much deep semantic information. At the same time, different weights are set according to the importance of different input features to ensure that the network can adjust the weight parameters adaptively during gradient backhaul, and pay attention to the importance of different features and adjust them. This innovation is the second improvement described in this article, AWFP-Add, and the formula and structure are shown in Fig. 4:

Figure 4 Feature network design.

(A) FPN (B) PAN (C) AWFP.

(1) O= ∑iwiɛ+ ∑jwj⋅Ii

(2) ki= Sigmoidwi

(3) Z=1−k∗y+k∗x.

Equation (1) is the mathematical expression of fast normalized fusion, for this study, directly introducing fast normalized fusion can improve model accuracy to a certain extent, but the improvement is very limited. Moreover, it introduces more learnable parameters, not only increasing the model’s computational complexity but also significantly affecting the model’s stability. For this reason, we opt to adopt the simple and efficient sigmoid function to restrict the weight value range between (0, 1) and implement the task of the network to focus on the importance of different features by itself through Eq. (3), and its complete computational process is illustrated in Fig. 5.

Figure 5 Add module design.

(A) The computational flow of the ordinary Add module. (B) The computational flow of Add module with learnable parameters.

Focal-SIoU

The loss function of YOLOv7-tiny consists of three components: bounding box regression loss (BBR), confidence loss function, and class loss function. The bounding box regression loss measures the error in the predicted box’s coordinate localization error. The confidence loss reflects the confidence error of the predicted box, while the class loss function captures the prediction box’s error in predicting the target class. Specifically, the class loss function uses binary cross-entropy (BCE) loss, only calculating the classification loss of positive samples. The confidence loss function is also BCE loss, but it measures the confidence loss between the predicted bounding box and the ground truth box using complete intersection over union (CIOU). This loss is calculated for all samples. The bounding box loss function also uses CIOU loss, but it only calculates the position loss of positive samples. The CIOU loss takes into account three important geometric factors: overlap area, the distance between centroid points, and aspect ratio, which makes the bounding box regression more stable. This is shown in Fig. 6. Given the predicted box B and the ground truth box Bgt. The CIOU loss is defined by Eq. (4): (4) LossCIOU=1−IOU+ρ2b,bgtc2+αv

Figure 6 CIOU figurative expression form.

where ρ2b,bgt denotes the Euclidean distance between the center point of the prediction box and the center point of the ground truth box, denoted by d in Fig. 6, c denotes the diagonal distance between the ground truth box and the smallest closed rectangle contained in the prediction box. α is defined by Eq. (5) below and ν is defined by Eq. (6) below.

(5) α=v1−IOU+v

(6) v=4π2arctanwgthgt− arctanwh2

where wgt represents the width of the ground truth box, hgt represents the height of the ground truth box, w represents the width of the prediction box, h represents the height of the prediction box. However, there is still room for improvement in the boundary box loss function. For instance, the CIOU does not take into account the orientation between the ground truth and predicted bounding boxes, which results in slower convergence. To address this, we propose replacing the original CIOU loss function with scylla intersection over union (SIOU), which introduces the vector angle between the ground truth and predicted bounding boxes to redefine correlation (Gevorgyan, 2022). The SIOU loss function consists of four components: distance loss, angle loss, shape loss, and IOU loss.

(1) ANGLE COST

The angle cost is defined by Eq. (7). Its diagram is shown in Fig. 7. (7) Λ=1−2×sin2arcsinchσ−π4

Figure 7 Angle cost.

where ch is the height distance between the center point of the ground truth box and the prediction box, σ is the distance of the center point between the ground truth box and the prediction box.

(8) ch= maxbcygt,bcy− minbcygt,bcy

(9) σ=bcxgt−bcx2+bcygt−bcy2

where bcxgt,bcygt is the center coordinate of the ground truth box and bcx,bcy is the center coordinate of the prediction box.

(2) DISTANCE COST

The distance cost is defined by Eq. (10).Its diagram is shown in Fig. 8.

Figure 8 Distance cost.

(10) Δ= ∑t=x,y1−e−γpt=2−e−γρx−e−γpy

(11) ρx=bcxgt−bcxcw2,ρy=bcygt−bcych2

(12) γ=2−Λ

where (cw, ch) is the width and height of the minimum external matrix of the ground truth box and the prediction box.

(3) SHAPE COST

The shape cost is defined by Eq. (13).

(13) Ω= ∑t=w2h1−e−wtθ=1−e−wwθ+1−e−whθ

(14) ww=w−wgt maxw,wgt,wh=h−hgt maxh,hgt

where (w, h) is the width and height of the prediction box, wgt,hgt is the width and height of the ground truth box, θ controls the degree of attention to shape loss.

(4) IoU COST

The IoU cost is defined by Eq. (15). (15) IoU=AB

where A represents the intersection of the ground truth box and the prediction box, B represents the union of the ground truth box and the prediction box.

(5) SIoU LOSS

In conclusion, the SIoU loss is defined by Eq. (16). (16) LossSIOU=1−IOU+Δ+Ω2.

In boundary box regression, the problem of imbalanced training samples also arises, where the sparsity of the target objects in the images leads to a scarcity of high-quality examples with small regression errors compared to low-quality examples. To concentrate the SIOU loss on high-quality examples, this study considers combining the focal loss, which specifically deals with the imbalance between positive and negative samples, with the SIOU loss (Zhang et al., 2022). Therefore, we propose the Focal-SIOU loss to enhance the performance of the SIOU loss, and it is defined by Eq. (17). (17) LFocal-SIOU=IOUγLSIOU.

Experiment and Result Analysis

In this section, the data set, evaluation metrics, comparison objects, and methods are described and the experimental results are analyzed to confirm the validity of the improved model.

Datasets

In our experiments, we used two popular public datasets to validate the utility of the proposed method, namely GC10-DET (see Fig. 9) and NEU-DET (see Fig. 10).

Figure 9 GC10-DET deep metal surface defect dataset defect types.

Figure source credit: GC10-DET database.

Figure 10 NEU-DET steel surface defect dataset defect types.

Figure source credit: NEU-DET database.

(1) GC10-DET

The GC10-DET dataset contains 2,257 images of steel surface defect detection in actual industrial production. It includes 10 defect categories: Pu (Punching_hole), Wl (Welding_line), Cg (Crescent_gap), Ws (Water_spot), Os (Oil_spot), Ss (Silk_spot), In (Inclusion), Rp (Rolled_pit), Cr (Crease), and Wf (Waist folding). The images have a resolution of 2048*1000. The dataset is divided into training, validation, and testing sets in an 8:1:1 ratio.

(2) NEU-DET

The NEU-DETdataset was produced by a Northeastern University team (He et al., 2019). It contains six types of defects: Rs (Rolled-the_scale), Pa (Patches), Cr (Crazing), Ps(Pitted_surface), In (Inclusion), and Sc (Scratches). There are a total of 1,800 images in the dataset. The image resolution is 200*200. All images are grayscale. The training, validation, and test sets are divided into 8:1:1 ratios.

Experimental setup

We use the PyTorch deep learning framework to train and test our proposed model. The experimental setup consists of an AMD 15vCPU, RTXA5000 GPU, 24GB of memory, and the SGD optimizer for model optimization. A larger batch size is beneficial as it improves the model’s detection performance. Therefore, in this study, we use a batch size of 32 and train for 500 epochs with image sizes set to 640 × 640. During the training process, we employ data augmentation techniques such as random flipping, contrast adjustment, cropping, and scaling transformations to enhance the model’s robustness.

Performance evaluation

In industrial production, the accuracy and speed of defect detection are two critical factors. Incorrect results in terms of defect type or location can lead to machine misjudgment. Slow inspection speeds significantly reduce the efficiency of defect detection and could even cause accidents. To address these issues, three measurement values, namely AP, mAP, and FPS, are used to evaluate the strip defect detection model. AP represents the average accuracy for each defect, mAP represents the average accuracy for all categories, and FPS represents the frames per second. These metrics are used to determine whether the model meets the requirements for real-time monitoring. The calculations for these metrics are as follows: (18) Precision=TPTP+FP

(19) Recall=TPTP+FN

(20) AP= ∫01PRdR

(21) mAP=∑i=1cAPic.

In the given context, TP represents the number of defect samples correctly detected, FP represents the number of non-defect samples detected, FN represents the number of defect samples falsely detected, and P and R represent precision and recall respectively.

Comparisons with other methods on GC10-DET

The experimental results on the metal surface defect dataset using the improved YOLOv7-tiny model are shown in Table 1. The detection results, illustrated in Fig. 11, display each predicted defect area enclosed in a box, along with the corresponding defect category and confidence level. As depicted in the figure, the improved model accurately locates and classifies defects, predicts their sizes, and exhibits relatively good detection performance even for small defects.

Table 1 Detection results of GC10-DET dataset.

Metrics	All	Wf	Pu	Wl	Cg	Ws	Os	Ss	In	Rp	Cr	
Precision (%)	86.4	100	90.4	91.1	91.7	84.2	75.7	85.2	75.2	80.4	90.0	
Recall (%)	74.9	77.4	95.0	37.2	92.9	83.1	84.7	69.7	69.1	70.8	69.2	
mAP@0.5 (%)	81.0	85.1	95.7	68.4	95.0	86.4	84.4	78.5	67.3	73.0	76.2	

Figure 11 Visualization of the detection results of GC10-DET dataset.

Figure source credit: GC10-DET database.

To evaluate the effectiveness of our proposed model, we compared our approach with some recently published and highly effective methods, as shown in Table 2. According to Table 2, our proposed model achieved the highest mAP and the fastest detection speed compared to other methods. It also outperformed in terms of accuracy on three commonly challenging defect categories (Os, Ss, In). The improved model in this study reached an mAP of 81, which is a 6.9 increase compared to the improved YOLOv5 mAP. Compared to the improved YOLOv3 model, it had a 9.7 higher mAP and outperformed YOLOXD with a 2.55 mAP advantage. In comparison to LFF-YOLO, the improved model in this study showed an almost 21 improvement in mAP. Compared to DCC-CenterNet, our model achieved a 19.07 increase in mAP and a 4.6 times improvement in FPS. Besides, compared to EDDN, our model showed significant improvements in mAP (from 65.10 to 81) and FPS (from 30.3 to 144.9). Although FANet achieved the highest accuracy in defect categories Wf, Pu, Cg, and Ws, its mAP was 0.5 lower than our proposed model, and its detection speed was far inferior. Compared to RDD-YOLO, our model increased the mAP from 75.2 to 81 and the FPS from 57.5 to 144.9. Finally, as shown in Fig. 12, it can be concluded that our proposed model achieved the best performance in terms of speed and accuracy.

Table 2 Detection results of state-of-the-art methods on GC10-DET.

Bold values indicate the maximum value of the comlum. The larger the value, the better the detection effect.

References	Methods	Wf	Pu	Wl	Cg	Ws	Os	Ss	In	Rp	Cr	mAP	FPS	
Kou et al. (2021)	YOLOv3	90.4	76.6	95.0	89.6	64.2	55.2	74.3	23.6	43.9	100	71.3	45.6	
Wang, Teng & Zou (2022b)	YOLOv5	43.1	95.7	85.0	95.3	79.0	65.4	65.3	43.0	74.1	54.2	74.1	–	
Wang et al. (2022a)	YOLOXD	91.1	99.3	96.7	96.7	84.1	73.2	75.4	38.1	65.9	63.9	78.45	–	
Qian et al. (2022)	LFF-YOLO	75.2	94.4	94.5	89.2	79.1	49.2	61.1	14.7	6.2	34.6	59.78	–	
Yasir & Ahn (2023)	YOLOv5	92.5	89.5	88.5	94.8	65.4	62.2	75.0	35.1	36.7	62.1	70.18	–	
Yeung & Lam (2022)	FANet	94.7	99.9	95.7	99.4	88.7	65.0	74.4	53.9	67.7	65.9	80.5	37.1	
Lv et al. (2020)	EDDN	91.9	90	88.5	84.8	55.8	62.2	65.0	25.6	36.4	52.1	65.1	30.3	
Tian & Jia (2022)	DCC-CenterNet	76.5	84.1	85.5	96.1	77.3	50.8	54.7	30.1	13.9	49.9	61.93	31.47	
Liu et al. (2023)	MSC-DNet	84.0	95.5	96.1	94.9	76.5	66.5	65.8	34.1	53.4	48.5	71.6	–	
Zhao et al. (2023)	RDD-YOLO	89.4	96.8	95.8	98.6	87.2	69.2	70.1	35.9	49.5	58.9	75.2	57.5	
Ours,2023	YOLOv7-tiny	85.1	95.7	68.4	95.0	86.4	84.4	78.5	67.3	73.0	76.2	81.0	144.9	

Figure 12 Comparison with some of the latest target detection algorithms on GC10-DET.

Additionally, the detection accuracy and speed of the baseline model are elaborated in detail in the Table 5 (mAP: 70.2, FPS: 181.8). After considering the trade-off between two important metrics, mAP and FPS, our method achieves optimal performance.

Comparisons with other methods on NEU-DET

In order to further investigate the effectiveness and generalization performance of the improved model, experiments were conducted on NEU-DET, and the results are shown in Table 3. The detection results are illustrated in Fig. 13, and detailed comparisons with the latest methods can be found in Table 4. It is evident that our model achieves an mAP of 80.1 while maintaining a high detection speed. In comparison to the improved YOLOv3 model, our model shows significant differences in terms of mAP and accuracy in detecting most categories of defects. Our model’s mAP is 7.7 higher than EDDN, 2.6 higher than the improved YOLOv5 model, 0.87 higher than LFF-YOLO, 0.69 higher than DCC-CenterNet, and 0.7 higher than MSC-DNET. Additionally, our model demonstrates the best detection speed among all models listed in the table, surpassing the second-highest FPS by nearly 35 points. When compared to RDD-YOLO, our method remains competitive, with slightly lower accuracy but with an increased FPS from 57.8 to 106.4. Furthermore, Table 4 includes the FANet model, which has the highest mAP and the best accuracy in detecting defects across all categories. However, this model has a fatal drawback—its FPS is only 34, In practical production environments, the performance of devices can be influenced by external interferences, leading to fluctuations in the model’s detection speed. Consequently, the marginal difference between an achieved frame rate of 34 frames per second (fps34) and the industry-recognized real-time detection threshold of 30 fps is insufficient. It is evident that this level of performance falls short of achieving true real-time detection capabilities in an industrial context. Therefore, deployment in a genuine industrial setting is unfeasible. The above data indicates that our model is effective in detecting various defects while maintaining a fast detection speed.

Table 3 Detection results of NEU-DET dataset.

Metrics	All	Crazing	Inclusion	Patches	Pitted_surface	Rolled-in_scale	Scratches	
Precision (%)	80.8	70.7	77.2	85.4	89.6	74.3	87.5	
Recall (%)	74.2	46.9	82.1	86.3	76.1	64.8	88.8	
mAP@0.5 (%)	80.1	58.5	83.1	90.3	86.4	69.2	93.2	

Figure 13 Visualization of the detection results of NEU-DET dataset.

Figure source credit: NEU-DET database.

Table 4 Detection results of state-of-the-art methods on NEU-DET.

Bold values indicate the maximum value of the comlum. The larger the value, the better the detection effect.

References	Methods	Crazing	Inlcusion	Patches	Pitted _surface	Rolled -in scale	Scratches	mAP	FPS	
Kou et al. (2021)	YOLOv3	38.9	73.7	93.5	74.8	60.7	91.4	72.2	64.5	
Qian et al. (2022)	LFF-YOLO	45.1	85.4	94.5	86.3	67.7	96.1	79.23	63.24	
Yasir & Ahn (2023)	YOLOv5	59.1	86.7	95.1	85.9	65.6	72.6	77.5	–	
Yeung & Lam (2022)	FANet	62.9	96.9	95.8	98.5	83.4	98.2	89.3	34.0	
Lv et al. (2020)	EDDN	41.7	76.3	86.3	85.1	58.1	85.6	72.4	–	
Tian & Jia (2022)	DCC-CenterNet	45.7	85.1	90.5	85.4	76.7	95.8	79.41	71.37	
Liu et al. (2023)	MSC-DNet	42.4	84.5	94.3	91.5	71.6	92.0	79.4	–	
Zhao et al. (2023)	RDD-YOLO	52.9	85.9	94.4	86.2	70.7	96.6	81.1	57.8	
Ours,2023	YOLOv7-tiny	58.5	83.1	90.3	86.4	69.2	93.2	80.1	106.4	

Ablation study

Our experiment was conducted on two mentioned datasets, and we used ablation experiments to demonstrate the effectiveness of each improvement. To validate the effectiveness of each improvement on the YOLOv7-tiny network model, various combinations of improvement strategies were tested while controlling the variables.

1. YOLOv7-tiny with the DAC module is referred to as C-YOLO.

2. YOLOv7-tiny with the DAC module and the Focal-SIoU loss function is referred to as CF-YOLO.

3. YOLOv7-tiny with the DAC module and the AWFP-Add module is referred to as CA-YOLO.

4. YOLOv7-tiny with the DAC module, AWFP-Add module, and Focal-SIoU loss function is referred to as CAF-YOLO.

Experiments on GC10-DET

From Table 5, it is evident that our improvements have been effective. The original YOLOv7-tiny model (baseline) achieved an mAP of 70.2. In comparison, C-YOLO outperforms it with a 4.3 higher mAP. Furthermore, C-YOLO demonstrates superior detection accuracy than YOLOv7-tiny in all defect categories except for Pu. CF-YOLO and CA-YOLO achieve mAPs of 75.3 and 77.1 respectively, surpassing the original model by 5.1 mAP and 6.9 mAP. CAF-YOLO attains an mAP of 81%, the highest among them. Regarding specific defect types, CAF-YOLO boasts the highest performance in all defect detections except for Wl. The comparison of our improvement with the baseline model on the PR curve is shown in Fig. 14, confirming that our modification can identify various defects. CF-YOLO increases the mAP by +4.3 to +5.1, indicating that the mode of Focal Loss+SIoU to some extent mitigates the issue of imbalance between positive and negative examples in bounding box regression, as well as inaccurate regression boxes. By introducing learnable parameters in the Add module and incorporating the architecture of adaptive weighted fusion paths, the improvement range expands from +4.3 to +6.9, reflecting the positive impact of the AWFP-Add module.

Figure 14 Precision–recall(P-R)curves on GC10-DET.

(A)YOLOv7-tiny (baseline); (B) Improved YOLOv7-tiny (Ours).

Experiments on NEU-DET

To further investigate the effectiveness and robustness of our proposed method, we conducted ablation experiments on NEU-DET. Table 6 presents the results, showing that the baseline model YOLOv7-tiny achieved an mAP of 70.9. Meanwhile, C-YOLO, CF-YOLO, and CA-YOLO achieved mAP scores of 74.2, 76.3, and 76.7, respectively. The highest mAP is implemented by CAF-YOLO, and its comparison with the base model on the PR curve, as shown in Fig. 15, most accurately identifies defects such as Patches, Scratches, etc. The results indicate that when we use the DAC module, the accuracy of most types of defect detection increases. Nevertheless, the AWFP-Add module achieved a 2.5 mAP boost, and the use of Focal-SIoU resulted in a performance improvement of 2.1 mAP, which fully demonstrates the effectiveness of our modified Backbone and Head components.

Table 5 The Ablation experiments on GC10-DET.

Bold values indicate the maximum value of the comlum. The larger the value, the better the detection effect.

Methods	Wf	Pu	Wl	Cg	Ws	Os	Ss	In	Rp	Cr	Params (M)	GFLOPS	mAP	FPS	
YOLOv7-tiny	43.4	93.0	82.7	89.4	80.7	81.3	70.4	50.6	48.4	61.8	6.03M	13.1	70.2	181.8	
C-YOLO	48.6	92.7	83.9	91.7	84.4	82.7	73.4	54.8	61.3	71.4	11.06M	25.4	74.5	153.8	
CF-YOLO	64.5	93.0	83.9	93.0	83.7	83.1	74.4	59.8	59.3	57.8	11.06M	24.2	75.3	149.2	
CA-YOLO	74.3	93.3	84.3	92.2	84.3	83.1	74.1	59.0	56.8	69.3	11.86M	25.4	77.1	151.5	
CAF-YOLO	85.1	95.7	68.4	95.0	86.4	84.4	78.5	67.3	73.0	76.2	11.86M	25.5	81.0	144.9	

Table 6 The ablation experiments on NEU-DET.

Bold values indicate the maximum value of the comlum. The larger the value, the better the detection effect.

Methods	Crazing	Inclusion	Patches	Pitted _surface	Rolled -in_scale	Scratches	Params (M)	GFLOPS	mAP	FPS	
YOLOv7-tiny	37.7	74.9	89.3	81.4	53.3	89.8	6.02M	13.1	70.9	147.1	
C-YOLO	44.4	78.1	89.4	82.9	59.1	91.1	11.05M	24.2	74.2	119.1	
CF-YOLO	45.8	80.8	90.3	83.9	63.9	92.8	11.05M	24.2	76.3	119.1	
CA-YOLO	48.1	80.3	90.1	84.0	65.3	92.3	11.85M	25.4	76.7	113.6	
CAF-YOLO	58.5	83.1	90.3	86.4	69.2	93.2	11.85M	25.4	80.1	106.4	

Figure 15 Precision–recall (P-R) curves on NEU-DET.

(A) YOLOv7-tiny (baseline); (B) improved YOLOv7-tiny (ours).

Failure cases analysis

Although our improved YOLOv7-tiny model has demonstrated good defect detection performance on the GC10-DET dataset, instances of detection failure still occur. In this study, we analyze typical cases of detection failure, as illustrated in Fig. 16. The bounding box without a numeric value represents the true defect area, and the bounding box with a numeric value represents the inspection result.

Figure 16 Failure cases of defect detection.

(A, B) The case where different defect images are marked as 1 defect and the detection results is multiple defects. Figure source credit: GC10-DET database.

Based on the displayed detection results, we conclude that low contrast is one of the reasons for detection failure. In some cases, although the network accurately identifies defects, the boundaries of these defects are blurry, resulting in a single defect area being detected as two or more adjacent defects, or multiple adjacent defects being detected as one defect. To address such situations, optimization of the defect annotation information in the dataset itself is necessary to avoid unnecessary problems caused by the quality of dataset labeling. Additionally, enhancing the feature extraction capability of the network can alleviate boundary confusion. Moreover, YOLOv7-tiny also possesses the ability of instance segmentation, which can be used for pixel-level defect detection.

Conclusion

In this article, we propose a novel detector for defect detection on metal surfaces, which is based on the YOLOv7-tiny model, with improvements on the Backbone and Head sections. The improved YOLOv7-tiny model demonstrates excellent performance in terms of detection accuracy and speed. To enhance feature extraction capability and expand the receptive field, we replaced the ELAN-T module in the original model with the DAC module, which serves as the main component of the backbone. In the Head section, AWFP aims to deepen the network and combine with the parameterized add module to more fully integrate features from different scales in order to obtain rich semantic information and stronger representation. Finally, the Focal-SIoU loss function is employed to better address issues such as inaccurate regression boxes. Experiments were conducted on GC10-DET and NEU-DET to validate the robustness and generalization of the proposed model. Compared to the state-of-the-art methods, the improved YOLOv7-tiny model achieved 81.0 mAP and 144.9 FPS on GC10-DET, and 80.1 mAP and 106.4 FPS on NEU-DET. These results sufficiently demonstrate the competitiveness of our proposed model among various metal surface defect detectors. All of these experimental results indicate that the model proposed in this paper meets the requirements of high detection accuracy and real-time detection.

However, for some obscure and minor defects, the performance of the model in this paper needs further improvement, and in the next study, we will further improve the network structure, such as the attention mechanism and some powerful feature extraction methods.

Supplemental Information

Supplemental Information 1 Code

The following section of improved code is included in this file: 1. The innovative DAC module in this article (in the yaml file) 2. The AWFP-Add module innovated in this article (in the yaml file), and the defined function (in common.py) 3. Improved Focal-SIoU loss function (in los.py under utils)

Additional Information and Declarations

Competing Interests

Author Contributions

Data Availability

The authors declare there are no competing interests.

Shuaiting Chen conceived and designed the experiments, performed the experiments, analyzed the data, performed the computation work, prepared figures and/or tables, authored or reviewed drafts of the article, and approved the final draft.

Feng Zhou analyzed the data, prepared figures and/or tables, authored or reviewed drafts of the article, and approved the final draft.

Gan Gao conceived and designed the experiments, performed the experiments, analyzed the data, performed the computation work, authored or reviewed drafts of the article, and approved the final draft.

Xiaole Ge performed the experiments, authored or reviewed drafts of the article, and approved the final draft.

Rugang Wang analyzed the data, authored or reviewed drafts of the article, and approved the final draft.

The following information was supplied regarding data availability:

The code and data for this article are available at Zenodo:

Shuaiting Chen. (2023). source code and raw data. https://zenodo.org/record/8219708.

The GC10-DET dataset is available at Kaggle: https://www.kaggle.com/datasets/alex000kim/gc10det.

The NEU-DET dataset is available at the NEU surface defect database homepage and Kaggle:

-http://faculty.neu.edu.cn/songkechen/zh_CN/zdylm/263270/list/index.htm

-https://www.kaggle.com/datasets/zy12345/neudet?resource=download.

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
