# Peer review of "Unleashing the power of AI in detecting metal surface defects: an optimized YOLOv7-tiny model approach"

_PeerJ Computer Science, doi:10.7717/peerj-cs.1727_

## Round 0.1 · original submission · Major Revisions

Based on the reviewer recommendations that appreciate the work and also suggest improvements, the authors should prepare a major revision addressing all comments.

**Language Note:** The review process has identified that the English language must be improved. PeerJ can provide language editing services - please contact us at copyediting@peerj.com for pricing (be sure to provide your manuscript number and title). Alternatively, you should make your own arrangements to improve the language quality and provide details in your response letter. – PeerJ Staff

Reviewer 1 ·

Basic reporting

The study presents original work to detect defects on metal surface using YOLOv7-tiny. In particular, it aims to improve defect detection by modifying the Backbone and Head sections of the YOLO architecture.

Experimental design

1. The paper is well-written and the experiments are convincing.

2. Why did you choose YOLOv7-tiny? Do you believe that the same results can be reproduced with other detection algorithms?

3. The authors need to better explain the context of this research, including why the research problem is important.

Validity of the findings

To improve the abstract, the proposed approach's performance improvements should be explicitly stated.

Additional comments

1. Abbreviations should be written out in full on first use.
2. Provide literature to support the statement written in lines 184-185.
3. Unnecessary indentation in line 245.

Reviewer 2 ·

Basic reporting

The use of English in the manuscript is generally adequate, however, there are some expressions that are too colloquial, some typos, and a few sentences that are eighter to complex or that have small typos. Some examples are:
-Please, indicate the meaning of each acronym before its first use. Some examples include lines 15, 41, 42, 99, 110, etc. This is specially relevant in cases where the acronym is presented later in the text (DAC and ELAN are two examples).
-Lines 30 to 33 state twice the same facts.
-Incorrect use of capital letters on line 41 ("On the other hand, For traditional...").
-Incorrect use of capital letters on line 229 ("Sigmoid funtion To restrict...").
-Excessive use of "but" in lines 225 to 231. Please, try to reformulate the paragraph.
-In line ¿232?, CIOU is presented as (Intersection over Union), what does the "C" stands for?
-In line ¿232?, before equation 5, there is a missing space after "alpha".
-Section "Focal-SIoU" makes excessive and inconsistent use of the word "where" (relating the use of capital letters after a period).
-The acronym for SIOU is not indicated.
-In lines 288-289 it is stated that the model is compared to a "recently publisher and highly effective method" (singular) while Table 2 present a comparison among many methods.

The manuscript would benefit from a full revision of the text to make sure these and other small typos are corrected.


Figure 2 presents the architecture of the proposed detector. However, the naming of the components is not clear when information about it is provided. Lines 172-174 indicate that the ELAN-T structure has been replaced with the DAC module, however, this is not clearly visible in the image (Is this replacement indicated as the "C5New3" block?), please, try to improve the naming of each component in the image.

Figure 12 presents the comparison among the different models and the baseline. One needs to read up to line 330 to know where the baseline comes from. Please, add it to Table 2 or reformulate the text so it is clear what the baseline is.

Experimental design

The experiments and methods are well described.
However, further discussion is needed in relation with the following topics:

-In line 274 it is stated that "A larger bath size is beneficial as it improves the model's performance". May you provide a bibliographic reference that support that statement or elaborate further?

-Are data augmentation techniques applied to every image in the datasets or are a subset of the data augmented?

-Why have you decided to detect the metal defects using bounding boxes rather than other pixel-wise methods such as semantic segmentation?. In line 364 it is stated that Yolov7-tiny could be used for pixel-level defect detection.

Validity of the findings

The results presented by the authors are relevant and demonstrate a great performance of their method in relation to other state of the art models.

Further discussion is needed relating to:
-Lines 314 to 316 state that FANet model cannot be deployed in a real industrial environment because it process only 34 FPS. Usually, the threshold for considering a model "able to perform in real time" is set in 30 FPS. Why does industrial enviroments need to raise this threshold up to 60 fps?

-Given that, in both datasets, the proposed method perform over 100 FPS, would the model benefic from using the YOLOv7 architecture as a base instead of the YOLOv7-tiny model? Please, elaborate on the reasons to use the tiny version of YOLOv7 instead of the full model.

Additional comments

The manuscript "Unleashing the power of AI in detecting metal surface defects: an optimized YOLOv7-tiny model approach" presents a novel model for the detection of defects in metal surfaces.
The manuscript presents the modifications that have been implemented on top of the Yolov7-tiny model to adequate it to the metal defect detection datasets and the results of the model show that the modifications have been sucessful for the task.
However, this reviewer considers that some changes need to be made to the manusctipt prior to its publication.

---

## Round 0.2 · accepted · Accept

The authors have addressed all the comments from round 1. The reviewers are satisfied with the changes.

Reviewer 1 ·

Basic reporting

No comment

Experimental design

No comment

Validity of the findings

No Comment

Additional comments

All concerns have been addressed.

Reviewer 2 ·

Basic reporting

Authors have addressed all my comments and have modified the manuscript accordingly.
I have no further comments

Experimental design

Authors have addressed all my comments and have modified the manuscript accordingly.
I have no further comments

Validity of the findings

Authors have addressed all my comments and have modified the manuscript accordingly.
I have no further comments

Additional comments

Authors have addressed all my comments and have modified the manuscript accordingly.
I have no further comments